# Reactivations of Latent Viral Infections Are Associated with an Increased Thr389 p70S6k Phosphorylation in Peripheral Lymphocytes of Renal Transplant Recipients

**DOI:** 10.3390/v13030424

**Published:** 2021-03-06

**Authors:** Maxim Cherneha, Johannes Korth, Meike Kaulfuß, Mirko Trilling, Marek Widera, Hana Rohn, Sebastian Dolff, Nina Babel, André Hoerning, Andreas Kribben, Oliver Witzke

**Affiliations:** 1West German Centre of Infectious Diseases, Department of Infectious Diseases, Universitätsmedizin Essen, University Duisburg-Essen, 45147 Essen, Germany; meike.kaulfuss@uk-essen.de (M.K.); Hana.Rohn@uk-essen.de (H.R.); Sebastian.Dolff@uk-essen.de (S.D.); Oliver.Witzke@uk-essen.de (O.W.); 2Department of Nephrology, Universitätsmedizin Essen, University Duisburg-Essen, 45147 Essen, Germany; johannes.korth@uk-essen.de (J.K.); Andreas.Kribben@uk-essen.de (A.K.); 3Institute for Virology, Universitätsmedizin Essen, University Duisburg-Essen, 45147 Essen, Germany; mirko.trilling@uni-due.de (M.T.); marek.widera@uni-due.de (M.W.); 4Medical Department I, University Hospital Marien Hospital Herne, Ruhr-University of Bochum, 44625 Bochum, Germany; nina.babel@charite.de; 5Department of Paediatrics and Adolescent Medicine, University Hospital Erlangen, 91054 Erlangen, Germany; andre.hoerning@uk-erlangen.de

**Keywords:** BK polyoma virus, PyVAN, human cytomegalovirus, p70 ribosomal S6 kinase, renal transplantation

## Abstract

Reactivations of BK polyoma virus (BKPyV) and human cytomegalovirus (HCMV) frequently cause life- and graft-threatening complications after renal transplantation. Both viruses are dependent on the mTOR pathway for replication. In this study we investigated the association of viral replication with mTOR activity in peripheral lymphocytes of renal transplant recipients. A flow-cytometry based assay for the measurement of Thr389 p70S6k phosphorylation, a surrogate marker of the mTOR pathway was established. Forty-eight adult renal transplant recipients were recruited to measure p70S6k activity in their peripheral blood mononuclear cells. This data set in conjunction with information concerning previous replication of BKPyV and HCMV was examined for correlations. Episodes of BKPyV replication were significantly associated with increased p70S6k phosphorylation in CD4^+^ T lymphocytes (*p* = 0.0002) and CD19^+^ B lymphocytes (*p* = 0.0073). HCMV infection of patients with a high-risk HCMV constellation of donor and recipient (D+/R−) was associated with increased p70S6k phosphorylation in CD19^+^ B lymphocytes (*p* = 0.0325). These associations were found to be independent of the trough levels of the immunosuppressive drugs. Conclusion: P70S6k phosphorylation in peripheral lymphocytes is associated with BKPyV reactivations and to a lesser extent with HCMV infections in renal transplant recipients.

## 1. Introduction

Due to advances in immunosuppression, especially through the action of calcineurin inhibitors (CNI), kidney transplantation has become the best treatment for end-stage renal disease with regard to long term survival and quality of life [1,2,3].

Nevertheless, the management of transplant recipients poses a challenge since immunosuppression for transplant acceptance has to be balanced with sufficient immune competence to prevent infections. A leading cause of infections after transplantation are reactivations of opportunistic viruses from pre-existing latency reservoirs [4,5].

After renal transplantation, HCMV and BKPyV are responsible for most viral complications, threatening the life of patients and compromising graft survival. BKPyV shows the highest prevalence of viremia and causes polyomavirus-associated nephropathy (PyVAN) in up to 10% of the patients, doubling the occurrence of graft loss [4,6,7,8,9,10].

Although management algorithms for HCMV have been established in recent years, the only relevant options for antiviral treatment and prevention of PyVAN remain the overall reduction of immunosuppression. Off-label use of cidofovir or leflunomide has not been shown to be beneficial [4,7,8,10,11,12].

Interestingly, the use of inhibitors of the mechanistic target of rapamycin (mTORi) is favorable in case of viral infection. It is associated with fewer incidences of BKPyV DNAemia and PyVAN compared to a CNI based regimen, especially if the latter is based on tacrolimus (TAC). [13,14,15] The risk for HCMV reactivation in renal transplant recipients is also significantly reduced in mTORi-based regimens [16,17].

Like all viruses, BKPyV and HCMV seize control of the host translational machinery in order to synthesize viral proteins. The mTOR-pathway is particularly important in this respect. mTOR, a 289-kDa serine/threonine kinase, is an essential regulator of cellular metabolism, cell growth and proliferation [18,19]. It is found in two distinct protein complexes—mTORC1 and mTORC2. The mTORC1 is directly inhibited by mTORis, while mTORC2 is insensitive to rapamycin (RAP), although its structure can be disrupted after long-term exposure [18].

The two best-characterized substrates for phosphorylation of mTORC1 are the p70 ribosomal S6 kinase (p70S6k) and the eukaryotic translation initiation factor 4E-binding protein 1 (4EBP1). In their phosphorylated states p70S6k and 4EBP1 control the cap-dependent mRNA translation, protein synthesis and cell cycle progression. [18,19,20].

In vitro studies showed that BKPyV and HCMV are both greatly dependent on the activation of the mTOR pathway and have evolved mechanisms to ensure mTOR activity [18,21,22,23,24]. Thus, we hypothesized that the activity of the mTOR pathway is associated with virus infections like BKPyV and HCMV in renal transplant recipients. To assess the overall activity of mTOR-associated metabolism p70S6k phosphorylation, a widely used surrogate marker of the mTOR pathway’s activity, was used. Although mTORC1 phosphorylates p70S6k at three different sites, Thr389 phosphorylation was examined as it best reflects mTOR pathway inactivation by mTOR inhibitors [25,26].

## 2. Materials and Methods

### 2.1. Patient Characteristics

Forty-eight renal transplant recipients with an average age of 53.8 ± 13.6 years were included in this study after obtaining informed consent in accordance with the Declaration of Helsinki (Table 1). An institutional review board approval was obtained (approval number 14-5741-BO). Exclusion criteria were transplantation in the past 3 months, chronic infection with HIV, hepatitis B, or C and clinical signs of acute bacterial infection.

The patients have been assigned into subgroups depending on the occurrence of BKPyV or HCMV replication in their medical history, the HCMV serological status of donor and recipient and the type of prescribed immunosuppression. Heparinized blood samples were collected from each patient at our outpatient clinic.

BKPyV replication was defined as BKPyV DNAemia detectable in patient blood by PCR with a threshold of 400 genome copies/mL using the RealStar BKV PCR Kit 1.0 (Altona Diagnostics, Hamburg, Germany) [27]. Eighteen of the 48 patients showed a history of BKPyV replication, six of whom showed a PyVAN in at least one renal transplant biopsy [8,28].

HCMV replication was defined DNAemia detected in the blood by PCR with a threshold of 40 genome copies/mL using the Abott RealTime CMV Assay (Abbott Diagnostics, Wiesbaden, Germany) [29].

The HCMV low-risk group was defined by seronegativity of donor and recipient (D−/R−). The intermediate-risk group was defined as a HCMV-infected recipient irrespective of the serological status of the donor (D−/R+, or D+/R+). The high-risk group was defined as an HCMV-negative recipient receiving an HCMV-positive organ (D+/R−).

Twenty-four patients in the cohort received a TAC-based maintenance immunosuppression, while the remaining 24 patients received an mTORi based regime. Laboratory measurements and urine analysis were performed by the laboratory service of the University Hospital Essen. TAC trough levels were measured by the laboratory service of the University Hospital Essen, using an Architect i1000SR immunoassay analyzer (Abbott, Wiesbaden, Germany). Measurements of Everolimus (EVR) trough levels were performed by an external laboratory using a Liquid chromatography–mass spectrometry system.

### 2.2. PBMC Isolation and Flow Cytometry

Peripheral blood mononuclear cells (PBMCs) were isolated from heparinized blood within 3 h after sample collection by density gradient centrifugation using Lymphoprep medium (Stemcell Technologies, Cologne, Germany). T-cell medium was prepared with Gibco RPMI-1640 medium containing 1%(*v/v*) fetal calf serum, 1%(*v/v*) non-essential amino acids, 100 IU/mL penicillin/100 µg/mL streptomycin (Thermo Fisher Scientific, Waltham, MA, USA) and 1 mmol/L sodium pyruvate (Cambrex, Charles City, IA, USA).

Two million PBMCs per sample were stored in T cell medium for 1 h at 37 °C in a 5% CO2 atmosphere. As positive control, PBMCs of a healthy control subject were stimulated with PMA (10 ng/mL) and ionomycin (2 µg/mL) (Sigma-Aldrich, Steinheim, Germany) for 1 h. PBMCs were fixed with pre-warmed BD Phosflow Fix buffer I (BD Biosciences, Heidelberg, Germany) for 15 min at 37 °C. Cells were stained with mouse-antihuman CD19-Pacific Blue J3-119 (Beckman Coulter, Krefeld, Germany) and mouse-antihuman CD4-PerCP RPA-T4 (BioLegend, Koblenz, Germany) antibodies for 30 min at room temperature in the dark. The cells were washed and permeabilized with BD Phosflow Perm/Wash buffer (BD Biosciences, Heidelberg, Germany).

For intracellular staining, a primary mouse-anti-p70S6k(Thr389)-monoclonal antibody (Cell Signaling Technology, Leiden, Netherlands) and a mouse-antihuman IgG2a-κ Isotype control antibody (MG2a-53) (BioLegend) were used, which were pre-coupled with a PE fluorochrome using the Zenon™ R-Phycoerythrin Mouse IgG2a Labelling Kit (ThermoFisher scientific, Waltham, MA, USA). The cells were subsequently stained with the freshly coupled antibody for 30 min at room temperature in the dark.

Validation of the assay was conducted by performing side-by-side measurements with a method previously established by Hoerning et al. [30] for a part of the cohort (*n* = 16). A significant positive correlation of the results of both assays was evident for both CD4^+^ T lymphocytes (Spearman’s correlation coefficient r = 0.8676; *p* < 0.0001) and CD19^+^ B lymphocytes (Spearman’s correlation coefficient r = 0.694; *p* = 0.0019).

### 2.3. Statistical Analysis

Flow cytometry was performed on a Navios Flow Cytometer (Beckman Coulter, Krefeld, Germany) and data were analyzed using Kaluza Analysis version 1.5a software (Beckman Coulter, Krefeld, Germany). The level of phosphorylated p70S6k was quantified using the following formula: MFIx = [(MFIp70S6K − MFIIgG2a)/MFIIgG2a].

Statistical analysis for the flow cytometry assays was performed using GraphPad Prism version V 6.01 software (GraphPad Software, La Jolla, CA, USA). A normal distribution of the measured values was not assumed. The median MFIx values were compared with the Mann–Whitney U test. Data are depicted as median with the respective interquartile range (IQR). Correlation analyses were performed using the Spearman’s test. A one-tailed *p*-value < 0.05 was considered as statistically significant.

## 3. Results

### 3.1. BKPyV Viremia Is Associated with Increased p70S6k Phosphorylation

P70S6k phosphorylation in peripheral lymphocytes of renal transplant recipients with and without previous episodes of BKPyV viremia were compared. Renal transplant recipients with a BKPyV reactivation prior to the measurement showed a significantly increased p70S6k phosphorylation in CD4^+^ T lymphocytes (*p* = 0.0002) and CD19^+^ B lymphocytes (*p* = 0.0073) (Figure 1a,b). The relative difference in medians of p70S6k phosphorylation between groups was more pronounced in CD4^+^ T lymphocytes as compared to CD19^+^ B lymphocytes. Furthermore, renal transplant recipients with a PyVAN showed a trend towards even higher levels of p70S6k phosphorylation in their CD4^+^ T and CD19^+^ B lymphocytes without reaching the predetermined significance level. Interestingly, one of the analysed patients with PyVAN did not show viremia at the time of measurement. A post hoc exclusion of this data-point would lead to a statistically significant difference in p70S6k phosphorylation between patients with PyVAN and those with BKPyV viremia in CD4^+^ T lymphocytes.

### 3.2. HCMV DNAemia in Renal Transplant Recipients with a High-Risk Constellation Is Associated with Increased p70S6k Phosphorylation

P70S6k phosphorylation in peripheral lymphocytes was compared between renal transplant recipients with and without a documented episode of HCMV DNAemia. Without taking the individual risk constellations into account, p70S6k phosphorylation of patients with documented HCMV DNAemia was not significantly different compared to patients without, neither in CD4^+^ T lymphocytes (*p* = 0.4764) nor in CD19^+^ B lymphocytes (*p* = 0.2793) (Figure 2a).

In the subgroup of intermediate-risk patients (D+/R+, or D−/R+), also no significant difference was detectable between patients with and without HCMV reactivation, neither in CD4^+^ T lymphocytes (*p* = 0.4222), nor in CD19^+^ B lymphocytes (*p* = 0.1716) (Figure 2b).

However, a statistically significant difference in p70S6k phosphorylation was evident in the analysis of CD19^+^ B lymphocytes of high-risk patients (D+/R−). In this subgroup, patients with a history of HCMV DNAemia showed an increased p70S6k phosphorylation (*p* = 0.0325) (Figure 2c). The same trend was observed in the analysis of CD4^+^ T lymphocytes of high-risk patients, the difference between medians however not being significant (*p* = 0.0898).

### 3.3. Immunosuppressant Trough Levels and p70S6k Phosphorylation

Trough levels of TAC and EVR were measured concurrently to p70S6k phosphorylation. No statistically significant correlation was found between EVR trough levels and p70S6k phosphorylation, neither in CD4^+^ T lymphocytes (*p =* 0.3271) (Figure 3a), nor in CD19^+^ B lymphocytes (*p =* 0.0617) when analyzing all patients receiving EVR (Figure 3b). When analyzing a small subgroup of 4 patients receiving EVR without any CNI surprisingly a positive correlation was found between EVR trough levels and p70S6k phosphorylation. However, the validity of these results must be questioned because of the small sample size (Appendix A).

Conversely to the analyses of EVR trough levels, a statistically significant inverse correlation of p70S6k phosphorylation with TAC trough levels became evident for both CD4^+^ T lymphocytes (*p =* 0.0089) and CD19^+^ B lymphocytes (*p =* 0.01049) (Figure 3c,d). This association was not found to be significant any longer, when analyzing only patients receiving a combination therapy of TAC+EVR, or those only receiving an mTORi-free treatment (Appendix A).

To assess whether TAC trough levels influenced the previous analyses of virus reactivation, median TAC trough levels between groups were compared (Figure 4). TAC trough levels were not significantly different between patients with and without BKPyV reactivation (*p =* 0.318). Nor were they significantly different between the subgroups in the HCMV analysis (*p =* 0.4242).

## 4. Discussion

Given the importance of the mTOR pathway for virus replication [21,22,24], it was an apparent approach to investigate whether its activity is associated with the risk of viral reactivations in renal transplant recipients. In this study Thr389 p70S6k phosphorylation was used as a surrogate marker for mTOR activity. PBMCs were deemed a reasonable target since they are used as a latency reservoir by both HCMV [6,31,32] and BKPyV [33,34].

Notably, associations between increased p70S6k phosphorylation and virus reactivation were found for both BKPyV and HCMV.

These findings are consistent with the effects of BKPyV on host-cells, since it induces an increase in the overall activity of the mTOR pathway [21,35]. BKPyV mainly activates upstream regulators of mTORC in the early stages of infection, particularly AKT. This leads to increased phosphorylation of mTOR and subsequently of p70S6k. [21,23,24] Thus, it is not surprising that mTORis were shown to inhibit BKPyV replication at clinically relevant concentrations in vitro in a dose-dependent manner, accompanied by reduced p70S6k phosphorylation [23,24,36].

The data for HCMV are more difficult to interpret since significant associations could only be found in B lymphocytes of high-risk patients. Similarly to BKPyV, mTOR activity is pivotal for replication of HCMV. Upstream of the mTORCs, HCMV-infected cells exhibit an increase in the activity of AKT through a variety of mechanisms [21,22,37,38]. Additionally, the HCMV protein pUL38 directly antagonizes the tuberous sclerosis complex (TSC1/2) which usually inhibits mTORC1-activation by Rheb-GTP [39]. Furthermore, infection with HCMV alters the properties of the mTORCs, leading to a change in RAP sensitivity and substrate specificity of both mTORCs [38]. HCMV seems to induce a sequestration of mTORC1 into a perinuclear compartment, corresponding to the cytoplasmic viral assembly compartment, in order to maintain and manipulate mTOR activity [38,40]. Despite this complexity of interactions, an association of replication and increased p70S6k phosphorylation in peripheral lymphocytes agrees with our understanding of HCMV’s influence on the mTOR pathway.

The level and type of immunosuppression were among the apparent confounders on our analyses. Similarly to previous clinical studies, only TAC but not EVR trough levels were negatively correlated with p70S6k phosphorylation in our cohort when analyzing all eligible patients [30]. Part of the data even suggested a positive correlation of EVR trough levels with p70S6k phosphorylation (Figure 3b and Figure A1c,d).

However, most patients in the mTORi group were co-medicated with TAC, impeding any direct attribution of changes in p70S6k phosphorylation to the mTORi. A dose-dependent inhibition of the effects of mTORis on p70S6k phosphorylation by TAC, likely by competing for FKB12 interaction, has been previously described [24]. The contradictory positive correlation of EVR with p70S6k phosphorylation in peripheral lymphocytes (Figure A1c,d) is surprising and any attempt for an explanation would have to be purely speculative. However, the complete contradiction to the well-researched mechanism of action for mTORi and the extremely small sample size of *n* = 4 suggests a statistical outlier as the most likely explanation.

Despite the statistically significant negative correlation of TAC with p70S6k phosphorylation, median TAC levels between the groups selected for our analyses of viral reactivations did not significantly differ, thus not confounding these results.

Several limitations need to be considered, like the retrospective nature of our analyses and the lack of a direct temporal link between viremia and the conducted measurements. Therefore, we can only hypothesize what the actual causal connections between the observed parameters are. Other than the described direct viral effects on the mTOR pathway, it is tempting to assume that the mTOR pathway of peripheral lymphocytes is also influenced by paracrine mechanisms during virus reactivation. For instance, virus infections have been shown to modulate the host immune responses by secretion of orthologues to human cytokines like IL-10, which would lead to a distinct increase in peripheral p70S6k phosphorylation [41,42].

In addition, an adaptive immunological response to reactivations of latent viral infections would have to involve an activation of the mTOR pathway in several lymphocyte populations for effective lymphocyte proliferation and pathogen suppression [43,44]. This could mean, that any viral infection causing a substantial increase in lymphocyte activity will trigger similar changes of p70S6k phosphorylation, making it a general indicator of virus-specific immune responses.

This certainly opens further questions as to the possible uses of these connections. If in fact infections with viruses like BKPyV and HCMV lead to measurable increase in p70S6k phosphorylation of peripheral lymphocyte populations, this method could be of use in transplant recipient monitoring, especially in respect to the difficult and invasive diagnosis of PyVAN by kidney biopsy. If on the other hand it turned out that patients with an increase in Thr389 p70S6k phosphorylation just happen to be more susceptible for virus reactivation, an algorithm for risk stratification based on simple cytometric measurements might be possible to establish.

However, before the possibility of such implementations, further investigations with prospective longitudinal studies are needed to clear up the possible causal connections of our findings.

## Figures and Tables

**Figure 1 viruses-13-00424-f001:**
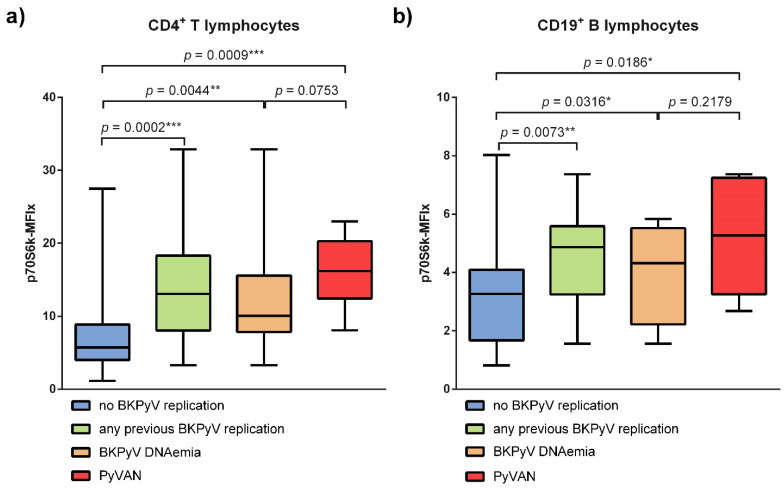
BKPyV reactivation is linked to increased p70S6k phosphorylation in peripheral lymphocytes (**a**) For CD4^+^ T lymphocytes the median p70S6k phosphorylation of patients without BKPyV reactivation was 5.73 (IQR = 4.02 to 8.88) and 13.07 (IQR = 8.05 to 18.30) for patients with a prior BKPyV reactivation (*p* = 0.0002). Renal transplant recipients with PyVAN, confirmed by kidney biopsy, (*n* = 6) showed a more pronounced increase in p70S6k phosphorylation with a median of 16.20 (IQR = 12.45 to 20.29) (*p* = 0.0009). The difference between patients with PyVAN and patients with only BKPyV DNAemia did not reach the predetermined significance level (*p* = 0.0753) with a median p70S6k phosphorylation of 10.07 (IQR = 7.84 to 15.59). (**b**) For CD19^+^ B lymphocytes the median p70S6k phosphorylation of patients without BKPyV reactivation was 3.27 (IQR = 1.67 to 4.09) and 4.87 (IQR = 3.25 to 5.59) for patients with prior BKPyV reactivation (*p* = 0.0073). Renal transplant recipients with PyVAN, confirmed by kidney biopsy, (*n* = 6) showed a more pronounced increase in p70S6k phosphorylation with a median of 5.27 (IQR = 3.25 to 7.24) (*p* = 0.0186). The difference between patients with PyVAN and patients with only BKPyV DNAemia did not reach the predetermined significance level (*p* = 0.2179) with a median p70S6k phosphorylation of 4.32 (IQR = 2.22 to 5.52). The bar charts show the medians with IQR of the phosphorylation of p70S6k. Mann–Whitney-U-Test was used to determine significance (* meaning *p* < 0.05; ** meaning *p* < 0.01; *** meaning *p* < 0.001).

**Figure 2 viruses-13-00424-f002:**
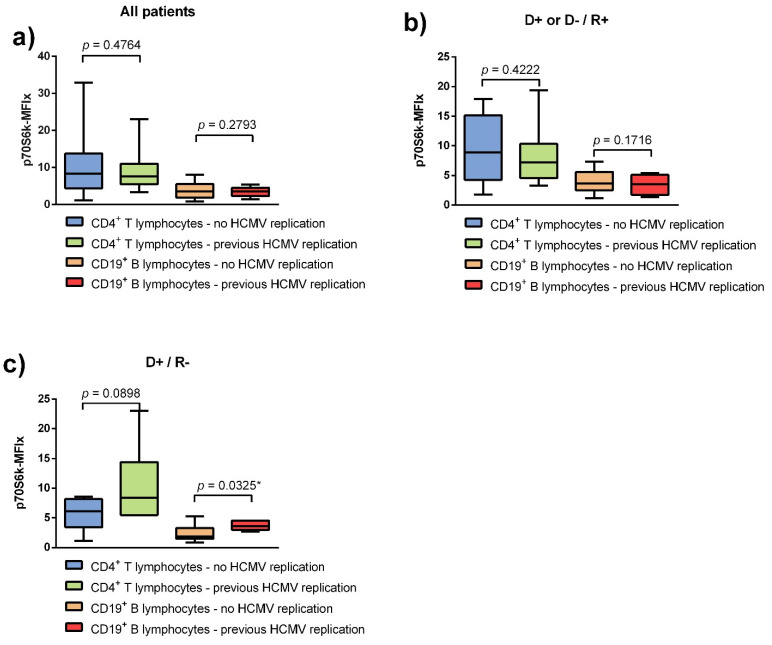
HCMV reactivation is only linked to increased p70S6k phosphorylation in renal transplant recipients with a high-risk constellation of donor and recipient. (**a**) Without differentiating between HCMV risk categories, no significant difference in p70S6k phosphorylation could be found between renal transplant recipients with (*n* = 16) and without (*n* = 32) prior episodes of HCMV DNAemia. For CD4^+^ T lymphocytes, the median p70S6k phosphorylation was 8.33 (IQR = 4.39 to 13.77) in patients without HCMV reactivation and 7.6 (IQR = 5.47 to 11.01) for patients with prior HCMV reactivation (*p* = 0.4764). For CD19^+^ B lymphocytes the median p70S6k phosphorylation was 3.54 (IQR = 1.84 to 5.52) in patients without HCMV reactivation and 3.53 (IQR = 2.30 to 4.56) for patients with prior HCMV reactivation (*p* = 0.2793). (**b**) For the subgroup of renal transplant recipients with an intermediate risk for HCMV no significant difference in p70S6k phosphorylation could be found between patients with (*n* = 10) and without (*n* = 11) prior reactivation of HCMV. For CD4^+^ T lymphocytes the median p70S6k phosphorylation was 8.87 (IQR = 4.25 to 15.16) in patients without HCMV reactivation and 7.21 (IQR = 4.56 to 10.35) for patients with prior HCMV reactivation (*p* = 0.4222). For CD19^+^ B lymphocytes the median p70S6k phosphorylation was 3.65 (IQR = 2.51 to 5.62) in patients without HCMV reactivation and 3.53 (IQR = 1.74 to 5.11) for patients with prior HCMV reactivation (*p* = 0.1716). (**c**) In the subgroup of renal transplant recipients with a high risk for HCMV, those with episodes of HCMV DNAemia in their medical history (*n* = 6) had a significantly increased p70S6k phosphorylation in CD19^+^ B lymphocytes but not in CD4^+^ T lymphocytes compared to patients without any history of HCMV reactivation (*n* = 6). For CD4^+^ T lymphocytes the median p70S6k phosphorylation was 6.13 (IQR = 3.43 to 8.15) in patients without HCMV DNAemia and 8.4 (IQR = 5.49 to 14.41) for patients with prior HCMV infection (*p* = 0.0.0898). For CD19^+^ B lymphocytes the median p70S6k phosphorylation was 1.85 (IQR = 1.47 to 3.28) in patients without HCMV DNAemia and 3 3.62 (IQR = 2.97 to 4.56) for patients with prior HCMV infection (*p* = 0.0325). The bar charts show the medians with IQR of the phosphorylation of p70S6k. Mann–Whitney-U-Test was used to determine significance (* *p* < 0.05).

**Figure 3 viruses-13-00424-f003:**
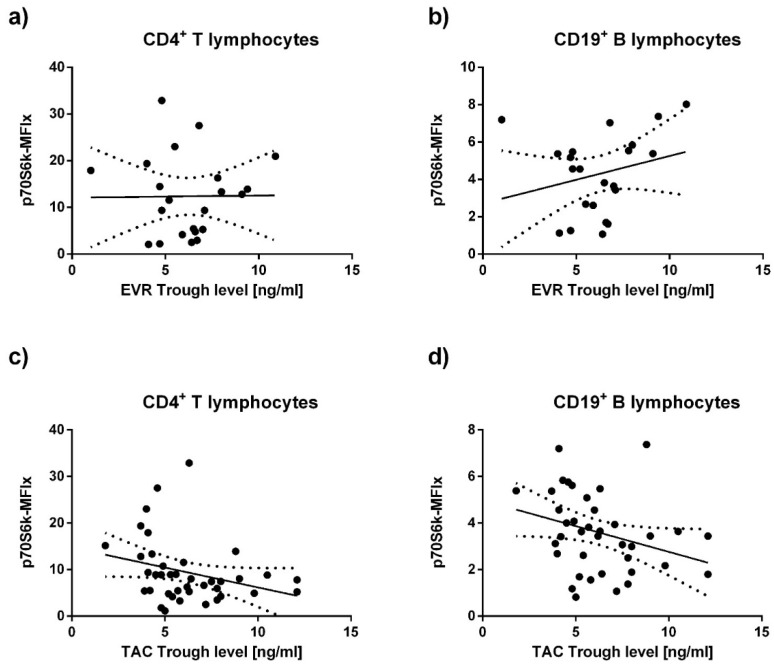
Tacrolimus but not Everolimus trough levels correlate with p70S6k phosphorylation. EVR trough levels at the day of measurement were not significantly correlated with p70S6k phosphorylation as measured in (**a**) CD4^+^ T lymphocytes (Spearman r = 0.1011, *p =* 0.3271) and (**b**) CD19^+^ B lymphocytes (Spearman r = 0.3384, *p* = 0.0617). TAC trough levels and p70S6k phosphorylation were found to negatively correlate when analyzing all patients of the cohort receiving TAC in both (**c**) CD4^+^ T lymphocytes (Spearman r = −0.3779, *p* = 0.0089) and (**d**) CD19^+^ B lymphocytes (Spearman r = −0.3689, *p* = 0.0104). Spearman’s rank test was used to determine statistical correlation.

**Figure 4 viruses-13-00424-f004:**
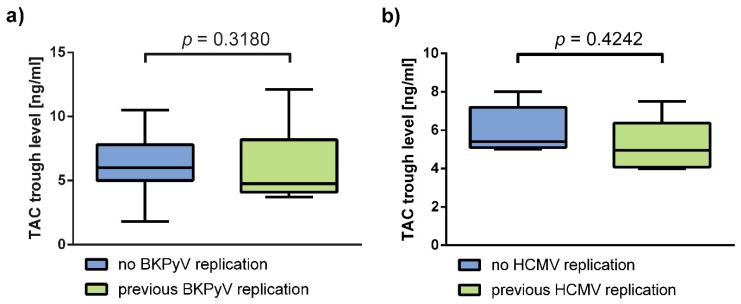
The compared subgroups show no significant difference in TAC trough levels. (**a**) TAC trough levels at the day of measurement were compared between patients with (*n* = 16, Median = 4.75, IQR = 4.10 to 8.2) and without (*n* = 23, Median = 6.00, IQR = 5.00 to 7.80) a previous occurrence of BKPyV reactivation. No significant difference in TAC trough levels was found between the two groups (*p* = 0.3180). (**b**) TAC trough levels at the day of measurement were compared between HCMV high-risk renal transplant recipients with (*n* = 6, Median = 4.95, IQR = 4.08 to 6.38) and without (*n* = 5, Median = 5.40, IQR = 5.10 to 7.20) a history of HCMV reactivation prior to measurement. No significant difference in TAC trough levels was found between the two groups (*p* = 0.4242). The bar charts show the medians with IQR of TAC trough levels at the day of the measurement. Mann–Whitney-U-Test was used to determine significance.

**Table 1 viruses-13-00424-t001:** Patients’ characteristics.

Variable	Measure	Overall (*n* = 48)		Female (*n* = 18)	Male (*n* = 30)
**Age [y]**	Mean (SD)	53.8 (13.6)		54.5 (10.7)	53.4 (15.2)
**Time since Transplantation [y]**	Median (IQR)	2.1 (1.0; 5.0)		1.7 (1.1; 4.5)	2.5 (0.8; 6.1)
**Serum-Creatinine at day of measurement [mg/dL]**	Mean (SD)	1.92 ± 1.34		1.90 ± 2.07	1.93 ± 0.63
**Immunosuppression**					
CNI-based	n	24		12	12
mTORi-based	n	24		6	18
EVR only	n	5		2	3
EVR+CNI	n	19		4	15
**Virus Replication**		**CNI-based**	**mTORi-Based**		
**HCMV**					
D−/R− without viremia	n	6	8	3	11
D−/R− with viremia	n	0	0	0	0
D+, or −/R+ without viremia	n	4	7	5	6
D+, or −/R+ with viremia	n	8	2	5	5
D+/R− without viremia	n	3	3	2	4
D+/R− with viremia	n	2	4	2	4
Active replication at day of measurement		0	0	0	0
Time since last reactivation [d]	Median (IQR)	167 (24; 411)		-	-
**BKPyV**					
No viremia		16	14	13	17
Viremia		8	10	5	13
PyVAN		1	5	1	5
PyVAN with viremia at day of measurement		0	5	1	4
Time since last reactivation [d]	Median (IQR)	263 (5; 462)		-	-

## Data Availability

The data presented in this study are available on request from the corresponding author. The data are not publicly available due to privacy regulations.

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
