# Peer review of "Reactivations of Latent Viral Infections Are Associated with an Increased Thr389 p70S6k Phosphorylation in Peripheral Lymphocytes of Renal Transplant Recipients"

_viruses, 2021, doi:10.3390/v13030424_

Round 1

Reviewer 1 Report

Cherneha et al investigated the correlation between the increased p70S6k phosphorylation and, HCMV and BKPyV viral infections in renal transplant recipients. The viruses, HCMV and BKPyV are known to cause serious complications after kidney transplantation which can result in compromising the graft and even threatening patients’ life. The authors conducted 48 patient study showing BKPyP infections are strongly associated with increased p70S6k phosphorylation. Similar correlation was also found for HCMV with lesser extent. This manuscript is well written, and data was clearly presented after a well-designed, carefully conducted study. Since HCMV and BKPyV infections can lead to measurable increase in p7S6k phosphorylation in peripheral lymphocytes of the recipients, this work can be developed as a method for monitoring kidney recipients as the authors suggested.

This work can be publishable in Viruses after minor revisions.

1.Page 1 of 13, (Abstract) Line 9…  BKPyV “repliaction” should be “replication”

2.Page 3 of 13, Line 8. Please include the full definition of EVR : Everolimus

3.Table 1 should be organized in a better way to present data. There are 30 males and 18 females in this study, coincidently under BKPyV section there are 30 patients with no viremia and 18 patients with viremia. Can authors present the data separately for male and female patients?

Reviewer 2 Report

The authors investigate phosphorylation of the p70S6k kinase as a means to predict recurrent of BK or CMV infection. They look to see if there is a correlation of p70S6k activation to the use of tacrolimus and/or everolimus in kidney transplant recipients. I believe the study is of real interest to the transplant community.

As I discuss below, the authors need to clarify the groups in regards to their Immunosuppressive treatment: tac only, EVR only, both.  More information may be deduced if the relative levels of each IS med is examined, especially when used in combination.  That is, does a patient on higher levels of EVR and lower levels of tac behave more like an EVR only patient, or does any addition of tac make this patient behave more like a primary tac patient?

The authors may also want to look at their 'positive' viral results as 'undetectable' vs 'low pos' (< 400 copies/ml for BK, < 40 for CMV) vs 'quantifiable" (> 400 for BK, > 40 for CMV). 

Page 2 of 13:  'BKPyV replication was defined as BKPyV DNAemia detectable in patient blood by PCR with a threshold of 400 genome copies/ml using the RealStar BKV PCR Kit 1.0 (Altona Diagnostics, Hamburg, Germany) [28].'  Although the threshold of 400 copies/ml was used for test reporting, if you looked at ‘undetectable’ vs ‘< 400 copies/ml’, did you any different results?

Same question for HCMV defined as 40 copies/ml threshold vs undetectable?

Table 1:  Please show HCMV and BKPyV bases on CNI vs mTORi based immunosuppression.

Page 7 of 13:  'Conversely, a statistically significant inverse correlation of p70S6k phosphorylation with TAC trough levels became evident for both CD4+ T lymphocytes (p = 0.0089) and CD19+ B lymphocytes (p = 0.01049) (Figure 3c and d). This association was not found to be significant any longer, when analysing only patients receiving a combination therapy of TAC+EVR, or those only receiving an mTORi-free treatment (data not shown).'

Figure 3: Please add the above information of Tac only vs EVR only vs Tac + EVR into the results.  Please show clearly patient groups who were on everolimus only, tacrolimus only, and both.  Please clarify same groupings for Figure 4 and for your discussion section.

Page 10 of 13: You note in your discussion that most patients on EVR were also on tac.  Please clarify this information in your Table 1 so the reader is knowledgeable of the immunosuppressive treatment prior to presenting results.  Although you may have most EVR patients also on tac, is there a trough tac level that has less effect on the ‘EVR’ findings?  That is, do EVR + Tac patients who have tac trough levels less than 4 or 5 ng/ml behave more like EVR only patients, etc.?  I would also

Discussion: 

The finding that the highest EVR trough levels demonstrate no correlation to p70S6k in CD4+ T lymphocytes and the higher levels tend to HIGHER levels of p70S6k in CD19+ B lymphocytes still requires more discussion.  Again, if you look at the few patients you have on EVR only (we don’t know how many of these patients you have), is there a different finding?

Do you have different results if you look at virus findings not as ‘< 400 copies/ml’ for BK and ‘< 40 copies/ml’ for HCMV, but as ‘undetectable’ vs ‘low pos’ vs ‘high pos’?  Clinically, low positive PCRs lead to high positive PCRs if you do not decrease immunosuppression or add anti-virals.  Perhaps the low positive results are hampering the differentiation between true positive and negative findings.

Round 2

Reviewer 2 Report

The authors did a solid job with suggested data review.